# Feeding Strategies of Two-Stage Fed-Batch Cultivation Processes for Microbial Lipid Production from Sugarcane Top Hydrolysate and Crude Glycerol by the Oleaginous Red Yeast *Rhodosporidiobolus fluvialis*

**DOI:** 10.3390/microorganisms8020151

**Published:** 2020-01-22

**Authors:** Rujiralai Poontawee, Savitree Limtong

**Affiliations:** 1Department of Biological Science, Faculty of Science and Technology, Huachiew Chalermprakiet University, Bangphli, Samutprakarn 10540, Thailand; r.poontawee@yahoo.co.th; 2Department of Microbiology, Faculty of Science, Kasetsart University, Chatuchak, Bangkok 10900, Thailand; 3Academy of Science, The Royal Society of Thailand, Bangkok 10300, Thailand

**Keywords:** lipids, oleaginous yeast, two-stage fed-batch cultivation, feeding strategy, sugarcane top hydrolysate, crude glycerol

## Abstract

Microbial lipids are able to produce from various raw materials including lignocellulosic biomass by the effective oleaginous microorganisms using different cultivation processes. This study aimed to enhance microbial lipid production from the low-cost substrates namely sugarcane top hydrolysate and crude glycerol by *Rhodosporidiobolus fluvialis* DMKU-SP314, using two-stage fed-batch cultivation with different feeding strategies in a 3 L stirred-tank fermenter. The effect of two feeding strategies of 147.5 g/L crude glycerol solution was evaluated including pulse feeding at different starting time points (48, 24, and 72 h after initiation of batch operation) and constant feeding at different dilution rates (0.012, 0.020, and 0.033 h^−1^). The maximum lipid concentration of 23.6 g/L and cell mass of 38.5 g/L were achieved when constant feeding was performed at the dilution rate of 0.012 h^−1^ after 48 h of batch operation, which represented 1.24-fold and 1.27-fold improvements in the lipid and cell mass concentration, respectively. Whereas, batch cultivation provided 19.1 g/L of lipids and 30.3 g/L of cell mass. The overall lipid productivity increased to 98.4 mg/L/d in the two-stage fed-batch cultivation. This demonstrated that the two-stage fed-batch cultivation with constant feeding strategy has the possibility to apply for large-scale production of lipids by yeast.

## 1. Introduction

Lipid produced by oleaginous microorganisms, including bacteria, yeast, filamentous fungi, and microalgae, has received a great deal of attention as a potential alternative source of fatty acids for several applications. In addition, used as biodiesel feedstocks [1], microbial lipids could be used as a promising feedstock for sustainable oleochemical production such as biolubricants [2], wax esters [3], olegels as food additives [4], and cocoa butter-like lipids [5]. Oleaginous microorganisms are considered as oleaginous species that can accumulate intracellular lipids greater than 20% of their cell mass. For example, *Nocardia globerula*, *Rhodococcus opacus,* and *Streptomyces coelicolor*, were reported as oleaginous bacteria. Oleaginous microalgae that were reported include, *Chlorella protothecoides*, *C. vulgaris*, and *Scenedesmus quadricauda*. The most common oleaginous yeasts were *Cryptococcus albidus*, *Lipomyces starkeyi*, *Rhodosporidium toruloides*, *Rhodotorula glutinis*, and *Yarrowia lipolytica*. Moreover, some species of filamentous fungi *viz. Aspergillus terreus*, *Mortierella alpina*, and *M. isabellina*, were reported to produce special lipids such as gamma-linolenic acid (GLA), arachidonic acid (ARA), eicosapentaenoic acid (EPA), and docosahexaenoic acid (DHA) [6,7]. Among such microorganisms, yeasts have been frequently applied in microbial lipid production because their characteristics are superior to those of other microorganisms: They offer high growth rates, high cellular lipid contents, and low risk of contamination that is easy to scale-up [8,9]. Moreover, the yeast cultivation technology for high cell and lipid content is well-developed [1,8,9] and the fatty acid composition of yeast lipids can be manipulated by modifying the culture conditions [10]. One of the most important advantages of using yeasts as lipid producers is their capability in the utilization of a wide variety of low-cost substrates, such as agricultural residues, industrial by-products, and other organic wastes [11,12]. Accordingly, there are now many reports of lipid production by oleaginous yeast using low-cost substrates such as lignocellulosic biomass (i.e., wheat straw, sugarcane bagasse, rice straw, rice hulls, corncob, corn stover, etc.) and crude glycerol [12].

The oleaginous red yeast, *Rhodosporidiobolus fluvialis* DMKU-SP314 was proved as a potential lipid producer [9,13]. This strain accumulated a high quantity of lipid and its fatty acid profile was similar to the profiles of plant oils, it mainly consists of oleic acid (36.6%), palmitic acid (30.6%), and linoleic acid (22.7%). It produced a high lipid content of 75% of dry cell mass (18.2 g/L lipid and 24.3 g/L cell mass) by batch cultivation in a 2 L stirred-tank fermenter using a mixture of sugarcane top hydrolysate and biodiesel-derived crude glycerol as substrates [9,13]. Sugarcane top is a lignocellulosic biomass, which is very low in cost and plentiful in Thailand because it is the main byproduct of the sugarcane industry and is usually simply left in the field after cane harvesting. Poontawee et al. [13] reported that this byproduct had to be potentially used as raw materials for lipid production by *R. fluvialis* DMKU-SP314. Moreover, crude glycerol, a byproduct of biodiesel production, could be used as co-substrate with hydrolysate of sugarcane top by this strain, which greatly reduces the overall cost of lipid production. In our previous study, the effect of nutritional factors (such as carbon and nitrogen sources) and environmental factors (such as cultivation temperature, aeration rate, and agitation speed) on lipid production by *R. fluvialis* DMKU-SP314 were optimized in batch cultivation [13]. However, other effective cultivation processes for lipid production by this strain including two-stage fed-batch cultivation has never been investigated.

The fed-batch cultivation process, which is also known as semi-batch cultivation can be carried out by intermittently or continuously adding the essential nutrients for cell growth or product formation to the culture vessel during the operation [14]. Consequently, the concentration of limiting nutrients in the culture broth can be kept at an optimal level by adjusting the feed rates [14,15]. This makes it possible to maximize the product concentration or product yield at the end of the cultivation [14]. Moreover, since the substrate concentration can be kept at a desirable level, the inhibitory effect of a high concentration of substrate can be avoided. The advantages of the fed-batch cultivation process are as follows: It shortens fermentation time, achieves high cell concentration, increases productivity, diminishes substrate inhibition or end-product inhibition, reduces the viscosity of the culture broth, reduces water loss by evaporation, and gives a higher dissolved oxygen rate [15,16]. Fed-batch fermentation processes have been widely applied in many fermentation processes to produce various products and microbial lipids. The fed-batch cultivation process is superior to conventional batch cultivation and has been proven effective to increase the cell concentration and cellular lipid content in oleaginous microorganisms [16,17]. Moreover, this process is used to avoid substrate inhibition by maintaining the concentration of substrates such as glucose, lignocellulosic-derived sugars, glycerol, etc. at a low level [16,17,18,19]. Therefore, there are now many reports about the application of fed-batch fermentation processes for microbial lipid production by various oleaginous yeasts [16,17,18,20]. The two-stage cultivation strategy has been developed to increase lipid accumulation by oleaginous microorganisms. In the first stage, the cells are cultivated in a nutrient-rich medium to promote cell propagation. Then, the cells are transferred to a carbon source solution without auxiliary nutrients to enhance lipid accumulation in the second stage [21]. The deficiency of nutrients, particularly nitrogen or phosphorous, discontinued the biosynthesis of protein and nucleic acids and so inhibited, cell proliferation. Nevertheless, the carbon source was more efficiently converted into lipids, leading to higher lipid productivity and yield [22]. The procedure of two-stage cultivation usually consists of the cultivation of cells in a nutrient-rich medium to obtain high cell density; then, cells are collected from the culture broth by centrifugation and resuspended in a carbon source solution, such as xylose solution [23], pure glycerol [22], and crude glycerol solution [21,22], without auxiliary nutrients. While such procedures are practical in shake-flask cultivation; they are difficult to perform in large-scale production as in a fermenter. To simplify the operation steps in the fermenter, the two-stage cultivation strategy together with fed-batch cultivation are applied, as a result, there is no need to separate cells from the culture broth by centrifugation. This reduces the number of steps in the operation and overall costs and also avoids the loss of cells during the cell separation step and reduces the risk of contamination. Two-stage fed-batch cultivation has been applied for lipid production by various oleaginous yeasts [16,18,19,20,24,25,26,27,28].

Therefore, the objective of this study was to assess the feasibility of the two-stage fed-batch cultivation for effective lipid production by *R. fluvialis* DMKU-SP314 using low-cost substrates. The process was investigated on the parameters feeding starting time points, feeding modes, and feeding solution concentration.

## 2. Materials and Methods

### 2.1. Yeast Strain and Growth Conditions

The oleaginous yeast strain, *R. fluvialis* DMKU-SP314, used in this study was obtained from the Yeast Research Laboratory of the Department of Microbiology, Kasetsart University (DMKU), Bangkok, Thailand. The primary stock culture was maintained at −80 °C in yeast extract-malt extract (YM) broth (10 g/L glucose, 5 g/L peptone, 3 g/L yeast extract, 3 g/L malt extract, and 15 g/L agar) supplemented with 20% (*v/v*) glycerol. The working stock was preserved on YM agar slants at 8 °C and sub-cultured monthly. All shake flask cultivation was performed at 28 °C on a rotary shaker (TAITEC Bio-Shaker BR-300LF, Saitama, Japan) at a speed of 150 rpm.

### 2.2. Raw Materials

The sugarcane top used in this study was obtained from a sugarcane field in Kamphaeng Saen District, Nakhon Pathom Province, Thailand, which mainly contained 42.7% (*w/w*) cellulose, 25.5% (*w/w*) hemicellulose, and 5.8% (*w/w*) lignin. The crude glycerol was obtained from the biodiesel production plant of the Thai Oleochemicals Company, Limited (Bangkok, Thailand), which contained 82.06% glycerol, 0.10% methanol, 3.52% ash, 5.43% non-glycerol organic matter, and 8.89% water. Soybean powder was purchased from Doi Kham Food Products Co., Ltd., Thailand, which contained 38.71% protein calculated to be the total nitrogen content of 7%.

### 2.3. Preparation of Sugarcane Top Hydrolysate (STH)

The sugarcane top was pretreated with an alkaline-hydrogen peroxide solution according to the method of Lui et al. [29]. The pretreated-sugarcane top was hydrolyzed by Accellerase^®^ 1500 enzyme (DuPont, Itasca, IL, USA) as described previously by Poontawee et al. [13]. The obtained sugarcane top hydrolysate (STH) contained 0.3 g/L of total nitrogen and approximately 31 g/L of total reducing sugars (TRS). The sugar composition of the STH was glucose, xylose, and other sugars in a ratio of 3:1:1 [13].

### 2.4. Effect of Crude Glycerol Concentration

The lipid production medium used in this study contained STH, 59 g/L crude glycerol, 0.9 g/L (NH_4_)_2_SO_4_, 0.21 g/L soybean powder, 0.4 g/L KH_2_PO_4_ and 2.0 g/L MgSO_4_·7H_2_O, and had an initial pH of 6.1 (adjusted with 1.0 M NaOH) [13]. To study the effect of crude glycerol concentration on growth and lipid production by *R. fluvialis* DMKU-SP314, shake flask cultivation was carried out in the lipid production medium supplemented with different crude glycerol concentrations of 30–120 g/L. The inoculum was prepared in YM broth incubated at 28 °C and 150 rpm on a rotary shaker for 24 h and then inoculated into 100 mL of production medium to achieve an initial optical density of 600 nm (OD_600_) of 1.0 ± 0.05 (the initial cell concentration was approximately 2.6 × 10^7^ cells/mL). The cultivation was performed on a rotary shaker at 28 °C and 150 rpm.

### 2.5. Batch Cultivation in a 3 L Fermenter

In this experiment, lipid production by *R. fluvialis* DMKU-SP314 was conducted in a 3 L stirred-tank jar fermenter (B.E. Marubishi, model MDFT-N-3L, Tokyo, Japan) with a working volume of 2 L. The inoculum was prepared by two successive cell propagations as described previously by Poontawee et al. [13]. In brief, a pre-inoculum of yeast was grown in 50 mL of YM broth in a 250 mL Erlenmeyer flask, incubated at 28 °C and 150 rpm for 24 h and then transferred to 100 mL of lipid production medium in a 500 mL Erlenmeyer flask to give an initial OD_600_ of 1.0 ± 0.05 and then incubated further for 24 h. Four flasks of the inoculum were inoculated to the fermenter containing 1.6 L of the lipid production medium at a rate of 20% (*v/v*), to obtain an initial cell concentration of approximately 7.9 × 10^7^ cells/mL [13]. The batch cultivation was conducted at 28 °C with the agitation speed and aeration rate set at 300 rpm and 3 vvm, respectively. The pH of the culture was uncontrolled; however, the pH levels were monitored throughout the cultivation using a pH sensor (405-DPAS-SC-K85/225, Mettler-Toledo, Zürich, Switzerland) while foaming was controlled by automatic addition of silicon-based antifoaming agent. The culture broth samples were taken at 24 h intervals to determine cell mass, lipids, total reducing sugars (TRS) residue, and crude glycerol residue. The experiments were performed in duplicate.

### 2.6. Two-Stage Fed-Batch Cultivation in a 3 L Fermenter

To enhance lipid production by *R. fluvialis* DMKU-SP314 by improvement of the cultivation process, two-stage fed-batch cultivation was applied. The cultivation was conducted in a 3 L stirred-tank jar fermenter (B.E. Marubishi, model MDFT-N-3L, Tokyo, Japan) with an initial working volume of 1.2 L and a final working volume of 2 L. The inoculum preparation, inoculation, and cultivation conditions were carried out in the same way as in batch cultivation (Section 2.5). The medium for the first-stage cultivation consisted of STH, 1.5 g/L (NH_4_)_2_SO_4_, 0.35 g/L soybean powder, 0.4 g/L KH_2_PO_4_, 2.0 g/L MgSO_4_·7H_2_O, and the initial pH was 6.1. The feeding solution consisted of crude glycerol, 0.4 g/L KH_2_PO_4_, and 2.0 g/L MgSO_4_·7H_2_O and the pH was 6.1. To evaluate an appropriate feeding strategy for the second-stage, different feeding modes of 800 mL of feeding solution were studied. The culture pH was uncontrolled during fermentation but monitored throughout the cultivation.

In the pulse feeding mode, experiment 1, after the batch operation had run for 48 h, a total of 800 mL of crude glycerol solution containing 147.5 g/L crude glycerol was fed into the fermenter within a short period (approximately 15 min) to make the total concentration of crude glycerol in the process 59 g/L. In experiments 2 and 3, the crude glycerol solution (147.5 g/L crude glycerol) was fed after the batch operation had run for 24 and 72 h, respectively. In the constant feeding mode (continuous feeding at a constant rate), experiments 4–6, crude glycerol solution (147.5 g/L crude glycerol) was fed by a peristaltic pump (BT50S Micrometer Speed-Variable peristaltic pump, Baoding Lead Fluid Technology Co., Ltd., China) at different dilution rates *viz.* low (0.012 h^−1^), medium (0.020 h^−1^), and high (0.033 h^−1^) dilution rates, equivalent to feed rates of 9.6, 16.0, and 26.4 mL/h, respectively, after the batch operation had run for 48 h. In experiment 7, 800 mL of crude glycerol solution containing 175 g/L crude glycerol was fed with a dilution rate of 0.012 h^−1^ at 48 h after the batch operation began, the total concentration of crude glycerol in the process being 70 g/L. The culture broth samples were taken at 24 h intervals to determine cell mass, lipids, TRS residue, and crude glycerol residue. The experiments were performed in duplicate.

### 2.7. Analytical Methods

The cell mass was determined gravimetrically and expressed as cell dry weight (CDW; g/L). Briefly, 5 mL of culture broth was centrifuged at 4000 rpm, for 5 min, washed twice with distilled water, transferred to pre-weighed aluminum foil cup, and then dried at 80 °C for 18–24 h before weight measurement was made.

Lipid was extracted by the method of Bligh and Dyer [30] using chloroform-methanol with slight modification as described by Poontawee et al. [13] and transmethylated by the method of Holub and Skeaff [31]. Fatty acid composition was analyzed by a capillary gas chromatograph equipped with a flame ionization detector (Shimazu 2010, Shizuoka, Japan) using a silica megabore capillary column (Durabond 225, J and W Scientific, Santa Clara, CA, USA) as described in our previous work [13]. Lipid concentration was calculated as the sum of all fatty acid quantities and expressed as grams of total fatty acids per liter of culture broth (g/L), and lipid content was expressed as a percentage of total fatty acids per gram of dry cell mass (% of dry cell mass). The TRS concentration was determined by the 3,5-dinitrosalicylic acid (DNS) method [32]. The glycerol concentration was determined by a glycerol assay kit (Sigma-Aldrich, USA). All analyses were performed in triplicate and the error bars represent standard deviation (SD) from the mean value.

### 2.8. Statistical Analysis

The statistical significance was evaluated by one-way analysis of variance (ANOVA) using IBM SPSS version 22 (SPSS, Cary, NC, USA) and the individual comparisons were obtained by Duncan’s multiple range test (DMRT). A value of *p* < 0.05 was considered to indicate a significant difference between treatments.

## 3. Results and Discussion

### 3.1. Effect of Crude Glycerol Concentration on Growth and Lipid Accumulation

The effect of crude glycerol concentration on growth and lipid accumulation of *R. fluvialis* DMKU-SP314 was investigated by the addition of crude glycerol at different concentrations in the range of 30–120 g/L to the lipid production medium, which contained STH and other nutrients [13]. The result in Figure 1A shows that in the medium containing a low concentration of crude glycerol (i.e., 30 g/L), the growth of *R. fluvialis* DMKU-SP314 increased rapidly in the first four days and then a maximum cell mass concentration of 15.9 ± 0.1 g/L was reached at day 5 of cultivation. However, after six days of cultivation, cell mass did not increase. Similar trends were observed in the media containing crude glycerol concentrations of 40, 50, and 60 g/L, in which cell mass reached the maximum values of 17.4 ± 0.6, 18.6 ± 0.3, 20.0 ± 0.3 g/L at seven, eight, and nine days of cultivation, respectively. This probably was a result of the exhaustion of glycerol in the medium, and consequently, the proliferation of yeast cells ceased. It is noteworthy that at low crude glycerol concentrations (i.e., 30 g/L), after reaching the maximum lipid concentration of 7.8 ± 0.1 g/L at seven days of cultivation, lipid concentrations notably decreased (Figure 1B). This phenomenon was also observed when *R. fluvialis* DMKU-SP314 was cultivated in media containing 40 and 50 g/L crude glycerol. The reason for this could be that carbon starvation leads to a breakdown of stored lipids or lipid degradation, which is termed “lipid turnover” [33]. Moreover, lipid turnover could be observed in Zygomycetes fungi cultivated in glycerol when glycerol was taken up at low rates, while in *Rhodotorula*, this occurred when the glycerol concentration was lower than the threshold level for conversion to lipids [19]. The highest cell mass and lipid concentration, which were 20.5 ± 0.6 and 15.9 ± 1.0 g/L, respectively, were achieved when *R. fluvialis* DMKU-SP314 was cultivated in the medium containing 70 g/L crude glycerol (Figure 1A,B). On the contrary, when the yeast was cultivated in the media containing higher concentrations of crude glycerol (>70 g/L), growth and lipid production noticeably decreased. As a result, the lowest cell mass and lipid concentration, which were 13.0 ± 0.7 and 5.6 ± 0.3 g/L, respectively, were obtained when *R. fluvialis* DMKU-SP314 was cultivated in the medium containing 120 g/L crude glycerol. This could be because a high concentration of crude glycerol contained a large amount of impurities which became toxic to yeast cells [10] and led to high osmotic stress that inhibited cell metabolic activities [34]. Moreover, a high concentration of crude glycerol increased the viscosity of the medium, which influenced oxygen transfer through the medium and obstructed the downstream processing [27]. Dissolved oxygen (DO) concentration is one of the most important factors affecting the growth and lipid accumulation in oleaginous microorganisms. Rakicka et al. [35] reported that in the chemostat culture, the increasing of glycerol concentration in the feeding medium resulted in a dramatically decrease of DO. The limitation of oxygen can be rate limiting in carbon metabolism and results in citric acid secretion in *Y. lipolytica* JMY4086. This corresponds to the results obtained from our previous investigation using the Box–Behnken design in which we found that the optimal concentration of crude glycerol in the medium was in the range of 48–85 g/L, but that the concentration should not exceed 80 g/L [13]. This was consistent with the amounts of crude glycerol residues in the culture medium as shown in Figure 2, in which the crude glycerol was almost exhausted in the medium at the end of cultivation for 240 h when added in the range of 30–70 g/L. On the contrary, when crude glycerol was added to the medium at a concentration higher than 80 g/L, it was not completely consumed by *R. fluvialis* DMKU-SP314, particularly, at 110 and 120 g/L crude glycerol; only 45.5% and 41.1%, respectively, were consumed. The analysis of fatty acid profiles revealed that the lipids produced had slightly different fatty acid compositions when cultivated in media containing different crude glycerol concentrations (Table 1). Lipids produced mainly consist of oleic acid (34.2–36.1%), palmitic acid (27.4–32.9%), and linoleic acid (15.9–18.2%). These fatty acids constituted an approximate 80% of the total fatty acid composition. Notably, palmitic acid (16:0) slightly decreased with the increase of crude glycerol concentration. On the other hand, stearic acid (18:0) increased with the increase of crude glycerol concentration. Moreover, myristic acid (14:0) appeared in small amounts when *R. fluvialis* DMKU-SP314 was cultivated in the medium containing 50–80 g/L of crude glycerol. Nevertheless, the proportion of saturated fatty acids and unsaturated fatty acids were slightly different among the different crude glycerol concentrations. Gao et al. [36] reported that no significant difference in fatty acids composition and content of lipids produced by *R. toruloides* 32489 cultivated in pure glycerol and crude glycerol, indicating that the presence of impurities in crude glycerol did not influence the fatty acid biosynthesis pathway. Consequently, the crude glycerol concentration of 70 g/L was selected for further investigation in two-stage fed-batch cultivation.

### 3.2. Batch Cultivation in a 3 L Fermenter

In fed-batch cultivation we planned to use a 3 L fermenter, therefore, batch cultivation for lipid production was performed in a 3 L stirred-tank fermenter with a 2 L working volume, in the same way as had been done in the 2 L stirred-tank fermenter used in the previous study [13]. The conditions used, derived from the previous study, were a temperature of 28 °C, an agitation speed of 300 rpm, an aeration rate of 2 vvm and 20% (*v/v*) inoculum. As shown in Figure 3, nearly all glycerol (approximately 59 g/L) was consumed after cultivation for 192 h and a cell mass of 30.3 g/L and a lipid concentration of 19.1 g/L were achieved (Figure 3 and Table 2). Remarkably, the cell mass and lipid concentration obtained in the cultivation in the 3 L stirred fermenter under nonoptimized condition were slightly higher than those achieved in the cultivation in the 2 L fermenter (cell mass of 24.3 g/L and 18.2 g/L of lipid) [13]. Therefore, these operating conditions could be applied in the next investigation, which was a two-stage fed-batch cultivation in a 3 L fermenter.

### 3.3. Two-Stage Fed-Batch Cultivation in a 3 L Fermenter

In fed-batch cultivation processes, substrate feeding is one of the most important factors influencing lipid production [37]. The use of appropriate substrate feeding mode led to efficient conversion of the carbon source to desired products. Moreover, the feeding time and the composition of the feeding solution affect the lipid productivity [35,37]. By modifying the feed rate, it is possible to control growth and lipid accumulation in oleaginous microorganisms [25,38]. In the present study, fed-batch cultivation with a pulse feeding mode was investigated first. Based on the results from batch cultivation, nearly all reducing sugars were consumed by *R. fluvialis* DMKU-SP314 after 48 h of cultivation and approximately 10 g/L TRS remained unconsumed at the end of cultivation. Therefore, in experiment 1, the substrate feeding was initiated when the level of TRS in the medium was lower than 10 g/L, which was 48 h after the beginning of batch operation. Although, the pulse feeding of crude glycerol solution without auxiliary nutrients could result in an increasing C/N ratio of the medium which favors lipid accumulation; however, it dilutes cell concentration in the fermenter. Karamerou and Webb [37] suggested that the minimizing feeding volume by increasing feeding solution concentration could diminish the dilution effect; nevertheless, it could be lower by high cell density. As seen in Figure 4A, after crude glycerol solution was fed, cell concentration in the fermenter was diluted approximately 1.8-fold; however, cell density in the fermenter recovered within 24–48 h. When crude glycerol solution (147.5 g/L) was fed at 48 and 24 h after batch operation had begun (experiments 1 and 2, respectively), the cell mass increased to 33.2 and 31.0 g/L, respectively, after 240 h of cultivation, compared with the batch cultivation (30.3 g/L) (Figure 4A). However, it could be observed that when the start of the fed-batch stage was delayed to 72 h (experiment 3), a lower cell mass and lipid concentration of 26.3 and 13.3 g/L, respectively, were obtained. It would be due to that cells experience nutrient starvation until crude glycerol was fed at 72 h [37]. It was noted that crude glycerol was not completely consumed by *R. fluvialis* DMKU-SP314 and approximately 40 g/L crude glycerol remained in culture broth throughout the cultivation (Figure 4E). Although, the cell mass obtained from fed-batch cultivation with the pulse feeding mode (experiments 1–2) were higher than that of batch cultivation; the lipid concentrations and lipid content were not increased (Figure 4B,C). This was probably due to pulse feeding mode favor cell growth than lipid accumulation of *R. fluvialis* DMKU-SP314. The feeding starting time point at 48 h after the beginning of batch operation gave the highest cell mass concentration. For further investigation in experiments 4–7, the feeding starting time point was set at 48 h after batch operation. Yen et al. [28] found that a slight delay of growth was observed in *Rh. glutinis* BCRC21418 after crude glycerol was pulse fed in a 50 L airlift fermenter. They suggested that the lag phase of growth would result from the high concentration of crude glycerol. Since the biodiesel derived-crude glycerol usually contains various potentially toxic substances, which would be inhibitive to cell growth when using high concentration feeding; therefore, cells require a longer adaption period at the high crude glycerol concentration [28]. Therefore, to avoid the accumulation of high crude glycerol concentration, the constant feeding mode of an equal amount of crude glycerol solution was applied. In experiments 4–6, fed-batch cultivation with a constant feeding mode was investigated by feeding crude glycerol solution at 48 h after the beginning of batch operation using a peristaltic pump by varying the dilution rates at low (0.012 h^−1^), medium (0.020 h^−1^), and high (0.033 h^−1^) and the results are shown in Figure 5. At the low dilution rate of 0.012 h^−1^, *R. fluvialis* DMKU-SP314 produced 38.5 g/L cell mass (equivalent to a cell mass productivity of 160.4 mg/L/h and a cell mass yield of 0.54 g/g carbon consumed) and 23.6 g/L of lipid (equivalent to a lipid productivity of 98.4 mg/L/h and a lipid yield of 0.33 g/g carbon consumed). Moreover, the maximum specific growth rate (*μ*_max_) was 0.06 h^−1^. However, at a high dilution rate of 0.033 h^−1^ a lower cell mass (28.8 g/L) and lipid concentration (16.0 g/L) were obtained when compared at low and medium dilution rates. This implied that a high dilution rate was not suited for lipid production by *R. fluvialis* DMKU-SP314. Signori et al. [39] applied different feeding rates (0.57, 0.45, and 0.32 mL/min) for lipid production from crude glycerol by *Lipomyces* DSM 70295. They found that the highest feeding rate (0.57 mL/min) led to a rapid culture dilution and resulted in a low glycerol consumption rate and biomass production. Whereas the lowest feeding rate (0.32 mL/min) resulted in higher biomass production, but prolonged the lipid production period, probably due to a slower transition from the growth phase to the lipid accumulation phase. The feeding rate of 0.45 mL/min could avoid an excessive culture dilution leading to high lipid accumulation. Anschau et al. [25] studied chemostat cultivation of *L. starkeyi* DSM 70296 at a dilution rate of 0.03 and 0.06 h^−1^ and reported that the yields of cell mass and lipid obtained at a high dilution rate of 0.06 h^−1^ were lower than those obtained at a low dilution rate of 0.03 h^−1^. The reason for this may be that the microbial cells need to remain within the chemostat for a period of time to consume the nutrients and convert them to lipids. Our result was in contrast with that of Gill et al. [40] who found that maximum lipid accumulation in yeast requires a dilution rate of one-third of the value of the maximum growth rate. It is noticeable that in the constant feeding modes the culture volumes at different dilution rates were different that would affect the mass transfer in the fermenters. Therefore, the fermentation parameters such as the DO concentration maybe not be the optimized condition. Zhang et al. [41] investigated the effect of DO concentration on lipid accumulation by *Trichosporon oleaginosus* ATCC20509 in a 15 L fermenter and found that a high DO concentration (50–60%) enhanced cell growth while maintaining a low DO level concentration (20–30%) during lipogenesis phase results in high lipid content. They suggested that lipid production was improved at the optimum DO concentration. The addition of glycerol provided sufficient substrate for cell growth and lipid production but did not enhance lipid accumulation in *T. oleaginosus* ATCC20509. Although, the lipid content was not increased, the product yields and productivities obtained from two-stage fed-batch cultivation were higher than those obtained in batch cultivation. This result corresponded to the report of Lorenz et al. [27] that using a two-stage fed-batch cultivation process for lipid production by *Rh. glutinis* Rh-00301 achieved a high cell mass of 106 g/L in a short cultivation time of 80 h and an increase in the lipid content to 63% of dry cell mass. Moreover, they found that glycerol formed when *Rh. glutinis* Rh-00301 was cultivated under high osmotic conditions, which resulted from a high concentration of carbon source in the medium. The formation of this undesired byproduct could be reduced by using constant feeding of a carbon source (glucose) compared with pulse feeding. Citric acid is another byproduct produced along with lipids under nitrogen-limited and carbon excess conditions. Karamerou et al. [42] suggested that the supplied large amounts of glycerol led it to convert to both lipid and citric acid, conversely, the continuous supply of glycerol channeled it to lipids. As a result, more citric acid secretion was observed in pulse feeding mode compared with constant feeding modes of *Rh. glutinis* CICC 31596. Moreover, in continuous feeding modes, the citric acid secretion was reduced by increasing the flow rate. They concluded that pulse and continuous feeding of the same total amount of nutrients, resulted in similar values of biomass (25 g/L) and lipid content (~40%). However, continuous feeding at higher rates (> 0.8 g/L/h) led to an increase in both biomass (30 g/L) and lipid content (53%). Therefore, continuous feeding was an efficient approach to enhance lipid production by *Rh. glutinis* CICC 31596. Consequently, the low dilution rate of 0.012 h^−1^ was selected for our further investigation.

Based on the results obtained from Section 3.1, in experiment 7 the concentration of crude glycerol in the feeding solution was increased to 175 g/L (the total concentration in the process was 70 g/L). The result is shown in Figure 6 and implied that the feeding of crude glycerol at a concentration of 175 g/L was not appropriate for lipid production by *R. fluvialis* DMKU-SP314 because the yeast cells were unable to completely consume the glycerol during the fed-batch phase. As a result, cell mass and lipid production were not increased. Many reports suggested that a lower substrate concentration was more favorable for growth and lipid accumulation in fed-batch cultivation than a higher concentration due to the effect of substrate inhibition [19,43].

The fermentation parameters of two-stage fed-batch cultivation with different feeding strategies are summarized in Table 2. The cell mass and lipid production of *R. fluvialis* DMKU-SP314 were compared to those of other oleaginous yeasts using fed-batch cultivation processes as shown in Table 3. The cell mass and lipid yields obtained from this study were relatively low when compared with yields when pure sugars (such as glucose and sucrose) were fed [20,27]. As mentioned in Section 3.1, glycerol is a substrate more slowly metabolized by yeast compared with pure sugars such as glucose. Moreover, crude glycerol is a complex substrate that contains various impurities that would affect growth and lipid accumulation by oleaginous microorganisms. Thus, to improve the cell mass and lipid productivities by shortening the cultivation time, the cultivation process needs further investigation.

The analysis of the fatty acid composition of the lipid produced by *R. fluvialis* DMKU-SP314 revealed that there were slightly different amounts of fatty acids in the lipid produced under different feeding strategies (Table 4). Oleic acid (32.7–35.7%) was a major fatty acid in total lipids, followed by palmitic acid (28.2–31.8%) and linoleic acid (13.5–17.3%), respectively. However, the lipids obtained contained high amounts of monounsaturated fatty acids, which are favorable for the application in biodiesel production.

## 4. Conclusions

The results obtained from our present study suggested that the feeding strategy of fed-batch cultivation had a strong influence on both biomass and lipid production by *R. fluvialis* DMKU-SP314. In our present study, lipid production by *R. fluvialis* DMKU-SP314 using two-stage fed-batch cultivation with constant feeding improved the lipid yield (0.33 g/g of carbon source consumed) and lipid productivity (98.4 mg/L/h) compared with batch cultivation (0.23 g/g of carbon source consumed and 79.6 mg/L/h, respectively). The constant feeding of 147.5 g/L crude glycerol solution at a dilution rate of 0.012 h^−1^ after 48 h of batch operation achieved the highest cell mass and lipid quantity of 38.5 and 23.6 g/L, respectively. The use of renewable low-cost raw materials together with an effective fermentation process to produce a high product yield is an important strategy for establishing an economically feasible and sustainable process for microbial lipid production. Based on our results, a two-stage fed-batch fermentation was employed to enhance the biomass and lipid production that could be applied for a large-scale lipid production by *R. fluvialis* DMKU-SP314. However, they required a long cultivation period, therefore the other effective fermentation processes (such as repeated fed-batch or continuous processes) should be investigated to increase the productivities and reduce the operation costs. Moreover, to optimize the process, other fermentation parameters, such as the effect of DO, pH, inoculum concentration, need to be further investigated.

## Figures and Tables

**Figure 1 microorganisms-08-00151-f001:**
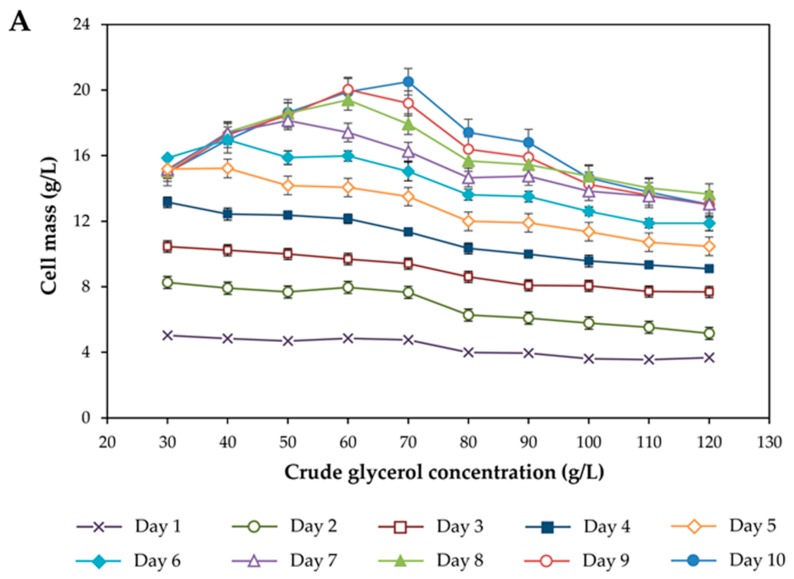
Effect of crude glycerol concentration on cell mass (**A**) and lipid (**B**) production by *Rhodosporidiobolus fluvialis* DMKU-SP314 in a shake flask culture at 28 °C and 150 rpm.

**Figure 2 microorganisms-08-00151-f002:**
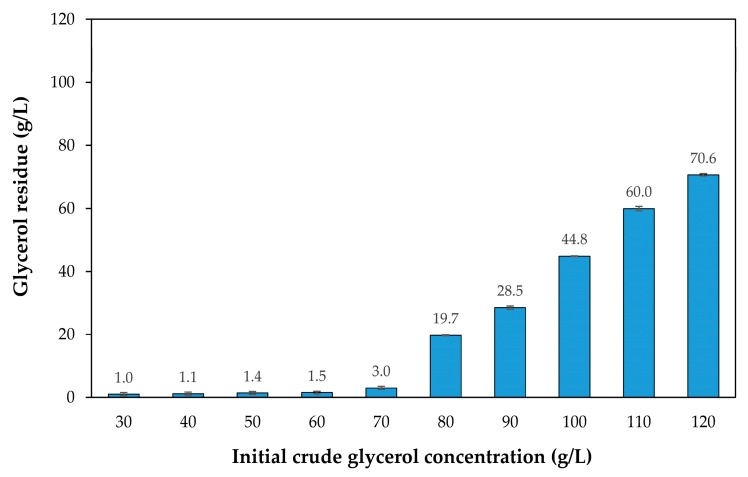
The amounts of glycerol residues in the medium containing different crude glycerol concentrations (30–120 g/L) after 240 h cultivation of *Rhodosporidiobolus fluvialis* DMKU-SP314 in a shake flask culture at 28 °C and 150 rpm.

**Figure 3 microorganisms-08-00151-f003:**
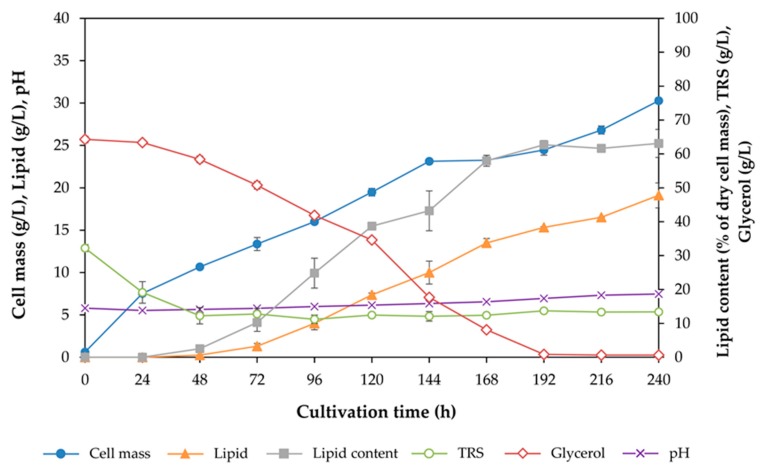
Time course of cell mass, lipid, lipid content, total reducing sugars (TRS), glycerol, and pH during batch cultivation of *Rhodosporidiobolus fluvialis* DMKU-SP314 in a 3 L stirred-tank fermenter at 28 °C, 200 rpm agitation speed, and 2 vvm aeration rate.

**Figure 4 microorganisms-08-00151-f004:**
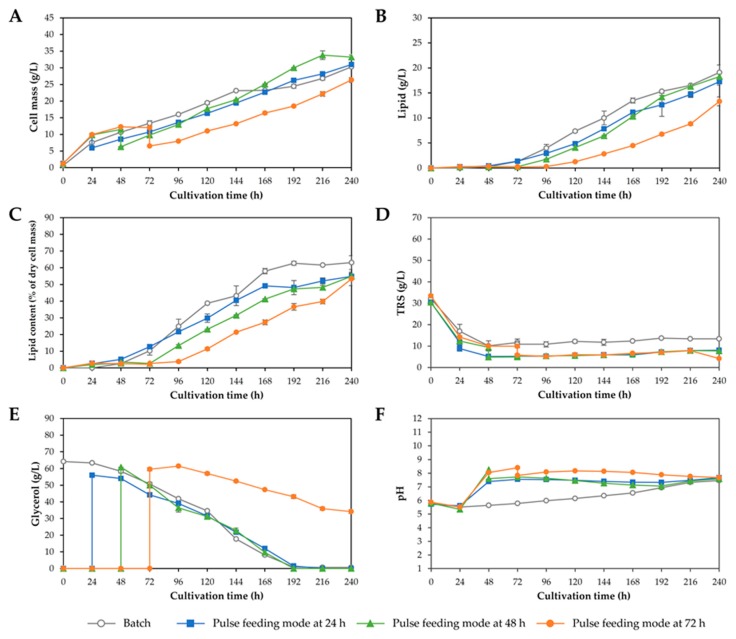
Lipid production by *Rhodosporidiobolus fluvialis* DMKU-SP314 in a 3 L stirred-tank fermenter using batch cultivation and two-stage fed-batch cultivation with pulse feeding modes at different starting time points of 24, 48, and 72 h after batch operation. (**A**) Cell mass, (**B**) lipid, (**C**) lipid content, (**D**) total reducing sugars (TRS), (**E**) glycerol, and (**F**) pH.

**Figure 5 microorganisms-08-00151-f005:**
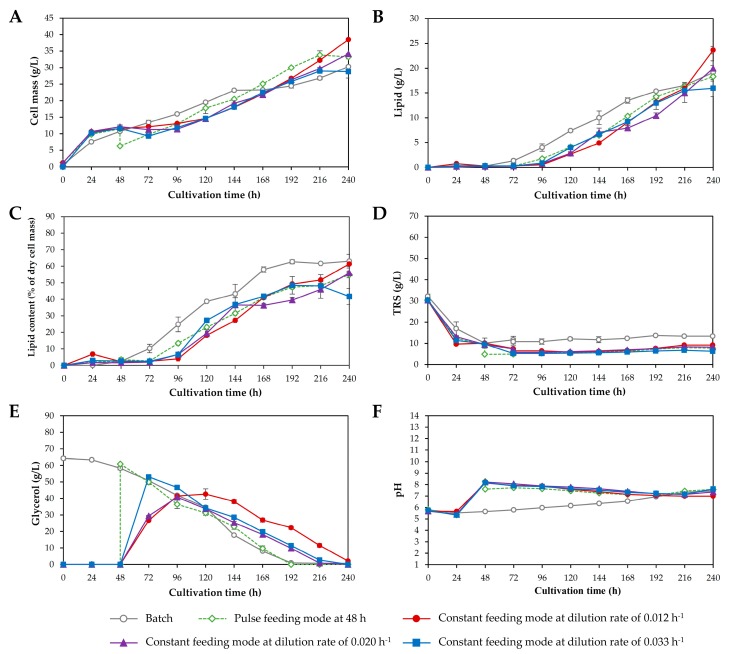
Lipid production by *Rhodosporidiobolus fluvialis* DMKU-SP314 in a 3 L stirred-tank fermenter using batch cultivation and two-stage fed-batch cultivation at a starting time point of 48 h with pulse feeding and constant feeding modes at different dilution rates of 0.012, 0.020, and 0.033 h^−1^. (**A**) Cell mass, (**B**) lipid, (**C**) lipid content, (**D**) total reducing sugars (TRS), (**E**) glycerol, and (**F**) pH.

**Figure 6 microorganisms-08-00151-f006:**
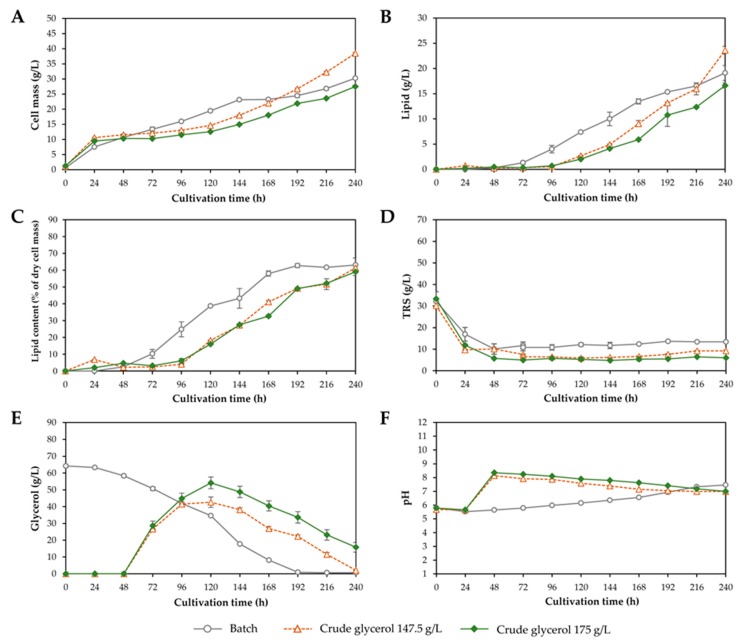
Lipid production by *Rhodosporidiobolus fluvialis* DMKU-SP314 in a 3 L stirred-tank fermenter using batch cultivation and two-stage fed-batch cultivation with constant feeding of different concentrations of crude glycerol solution of 147.5 and 175 g/L at the dilution rate of 0.012 h^−1^ after 48 h of batch cultivation. (**A**) Cell mass, (**B**) lipid, (**C**) lipid content, (**D**) total reducing sugars (TRS), (**E**) glycerol, and (**F**) pH.

**Table 1 microorganisms-08-00151-t001:** Fatty acid profiles of lipid produced by *Rhodosporidiobolus fluvialis* DMKU-SP314 cultivated in the sugarcane top hydrolysate (STH) medium containing different crude glycerol concentration (30–120 g/L) by shaking flask cultivation at 28 °C and 150 rpm for 240 h.

Fatty Acid	Fatty Acid (% of Total Fatty Acid) at Different Crude Glycerol Concentration (g/L)									
	30	40	50	60	70	80	90	100	110	120
Myristic acid (14:0)	nd	nd	0.5 ± 0.9 ^ab^	1.4 ± 0.1 ^a^	1.3 ± 0.0 ^a^	0.9 ± 0.8 ^ab^	nd	nd	nd	nd
Palmitic acid (16:0)	32.4 ± 0.0 ^a^	32.9 ± 0.4 ^a^	31.9 ± 0.9 ^a^	29.7 ± 0.9 ^b^	30.5 ± 0.8 ^b^	27.4 ± 1.4 ^c^	28.4 ± 0.3 ^bc^	28.7 ± 0.7 ^bc^	28.4 ± 0.0 ^bc^	27.4 ± 0.1 ^c^
Palmitoleic acid (16:1)	nd	nd	nd	nd	nd	nd	nd	nd	nd	nd
Stearic acid (18:0)	10.0 ± 0.8 ^f^	10.9 ± 0.2 ^e^	11.9 ± 0.5 ^d^	11.9 ± 0.3 ^d^	13.2 ± 0.1 ^c^	13.0 ± 0.6 ^c^	14.0 ± 0.2 ^b^	14.8 ± 0.4 ^a^	15.2 ± 0.1 ^a^	15.1 ± 0.2 ^a^
Oleic acid (18:1)	34.7 ± 0.4 ^a^	34.9 ± 0.2 ^a^	35.0 ± 0.3 ^a^	34.8 ± 0.4 ^a^	34.2 ± 0.2 ^a^	34.3 ± 1.8 ^a^	35.9 ± 0.5 ^a^	36.1 ± 0.2 ^a^	35.6 ± 0.1 ^a^	34.9 ± 0.6 ^a^
Linoleic acid (18:2)	18.2 ± 0.3 ^a^	17.6 ± 0.2 ^a^	16.9 ± 0.5 ^a^	18.1 ± 1.6 ^a^	15.9 ± 1.2 ^a^	18.1 ± 2.3 ^a^	17.4 ± 1.0 ^a^	16.5 ± 0.7 ^a^	16.9 ± 0.1 ^a^	17.5 ± 0.7 ^a^
γ-linolenic acid (18:3)	nd	nd	nd	nd	nd	nd	nd	nd	nd	nd
α-linolenic acid (18:3)	4.7 ± 0.0 ^ab^	3.7 ± 0.2 ^b^	3.8 ± 0.2 ^b^	4.2 ± 0.1 ^ab^	4.8 ± 0.6 ^ab^	6.4 ± 2.4 ^a^	4.3 ± 0.9 ^ab^	4.0 ± 0.3 ^ab^	4.0 ± 0.1 ^ab^	5.1 ± 1.3 ^ab^
Saturated fatty acids (SFAs)	42.4 ± 0.7 ^ab^	43.8 ± 0.6 ^ab^	44.4 ± 0.5 ^a^	42.9 ± 1.2 ^ab^	44.6 ± 0.7 ^a^	41.3 ± 2.6 ^b^	42.4 ± 0.6 ^ab^	43.5 ± 1.1 ^ab^	43.6 ± 0.1 ^ab^	42.6 ± 0.1 ^ab^
Unsaturated fatty acids (UFAs)	57.6 ± 0.7 ^ab^	56.2 ± 0.6 ^ab^	55.6 ± 0.5 ^b^	57.1 ± 1.2 ^ab^	55.4 ± 0.7 ^b^	58.7 ± 2.6 ^a^	57.6 ± 0.6 ^ab^	56.5 ± 1.1 ^ab^	56.4 ± 0.1 ^ab^	57.4 ± 0.1 ^ab^

nd: Not detectable. Values in the same row which have different superscript letters were significantly different (*p* < 0.05).

**Table 2 microorganisms-08-00151-t002:** Comparison of cell mass and lipid production by *Rhodosporidiobolus fluvialis* DMKU-SP314 using batch and two-stage fed-batch cultivations with different feeding strategies.

Response	Batch Cultivation	Two-Stage Fed-Batch Cultivation						
		1st	2nd	3rd	4th	5th	6th	7th
Initial working volume (L)	2.0	1.2	1.2	1.2	1.2	1.2	1.2	1.2
Final working volume (L)	2.0	2.0	2.0	2.0	2.0	2.0	2.0	2.0
Feeding starting time point (h)	-	48	24	72	48	48	48	48
Mode of feeding	-	Pulse	Pulse	Pulse	Constant	Constant	Constant	Constant
Feed rate (mL/h)	-	-	-	-	9.6	16.0	26.4	9.6
Dilution rate (h^−1^)	-	-	-	-	0.012	0.020	0.033	0.012
Concentration of crude glycerol in reservoir (g/L)	-	147.5	147.5	147.5	147.5	147.5	147.5	175
Total concentration of crude glycerol in the process (g/L)	-	59	59	59	59	59	59	70
Final pH medium	7.46 ^c^	7.58 ^b^	7.67 ^a^	7.68 ^a^	6.99 ^d^	7.36 ^c^	7.61 ^b^	7.00 ^d^
Cell mass concentration (g/L)	30.3 ^d^	33.2 ^c^	31.0 ^d^	26.3 ^g^	38.5 ^a^	34.2 ^b^	28.8 ^e^	27.4 ^f^
Lipid concentration (g/L)	19.1 ^bc^	18.3 ^cd^	17.3 ^de^	13.3 ^f^	23.6 ^a^	19.9 ^b^	16.0 ^e^	16.6 ^e^
Lipid content (% of dry cell mass)	63.1 ^a^	54.9 ^d^	54.8 ^d^	53.4 ^d^	61.4 ^ab^	55.9 ^cd^	41.6 ^e^	59.1 ^bc^
Cultivation time (h)	240	240	240	240	240	240	240	240
Maximum specific growth rate; *µ*_max_ (h^−1^)	0.104 ^a^	0.085 ^de^	0.085 ^e^	0.087 ^cd^	0.092 ^b^	0.092 ^b^	0.090 ^bc^	0.083 ^e^
Cell mass productivity; Q_X_ (mg/L/h)	126.2 ^d^	138.5 ^c^	129.1 ^d^	109.7 ^g^	160.4 ^a^	142.4 ^b^	120.0 ^e^	114.2 ^f^
Lipid productivity; Q_L_ (mg/L/h)	79.6 ^bc^	76.2 ^cd^	71.9 ^de^	55.5 ^f^	98.4 ^a^	83.1 ^b^	66.5 ^e^	69.3 ^e^
Total reducing sugar consumed (g/L)	20.0 ^c^	21.1 ^bc^	19.9 ^c^	23.6 ^ab^	20.5 ^c^	17.2 ^d^	18.9 ^cd^	25.4 ^a^
Crude glycerol consumed (g/L)	63.6 ^a^	60.8 ^b^	56.0 ^c^	59.6 ^b^	51.1 ^e^	53.0 ^d^	53.2 ^d^	37.7 ^f^
Cell mass yield (g/g) ^a^	0.36 ^e^	0.41 ^cd^	0.41 ^cd^	0.32 ^f^	0.54 ^a^	0.49 ^b^	0.40 ^d^	0.44 ^c^
Lipid yield (g/g) ^b^	0.23 ^c^	0.22 ^c^	0.23 ^c^	0.16 ^d^	0.33 ^a^	0.28 ^b^	0.22 ^c^	0.26 ^b^

Cultivation were performed in a 3 L stirred-tank fermenter at 28 °C, 200 rpm agitation speed, and 2 vvm aeration rate. ^a^ Gram of cell mass produced per gram of carbon source consumed. ^b^ Gram of lipid produced per gram of carbon source consumed. Values in the same row which have different superscript letters were significantly different (*p* < 0.05).

**Table 3 microorganisms-08-00151-t003:** Lipid production by various oleaginous yeasts using fed-batch cultivation.

Yeast Strain	Volume	Substrate	Cultivation Mode	Cultivation Time (h)	Cell Mass (g/L)	Lipid (g/L)	Cell Mass Productivity (g/L/h)	Lipid Productivity (mg/L/h)	Cell Mass Yield (g/g)	Lipid Yield (g/g)	Reference
*Rhodosporidiobolus fluvialis* DMKU-SP314	Fermenter (3 L)	STH and crude glycerol	Two-stage fed-batch (CF)	240	38.5	23.6	0.160	98.4	0.54	0.33	This study
			Two-stage fed-batch (PF)	240	33.2	18.3	0.139	76.2	0.41	0.22	
*Cryptococcus* sp. SM5S05	Flask (500 mL)	Glucose	Fed-batch (PF)	144	11.4	7.3	-	51	-	0.64	[16]
		Corncob hydrolysate	Fed-batch (PF)	144	10.8	6.6	-	46	-	0.61	
*Naganishia albida* KCTC 17541	Fermenter (1 L)	Onion waste and crude glycerol	Fed-batch (IF)	168	21.1	7.19	-	42.8	-	0.236	[26]
*Rhodotorula glutinis* CICC 31596	Fermenter (2 L)	Pure glycerol	Fed-batch (PF)	168	23	9.38	-	60	-	0.059	[42]
			Fed-batch (CF)	168	30.63	16.28	-	100	-	0.087	
*Cryptococcus albidus* var. *albidus* ATCC 10672	Fermenter (2.5 L)	Volatile fatty acids	Fed-batch (IF)	192	-	14.5	-	-	-	-	[24]
*Lipomyces starkeyi* DSM 70296	Fermenter (3 L)	Glucose:xylose (30:70)	Fed-batch (PF)	138	82.4	38.6	0.597	280	0.322	0.151	[25]
*Rhodotorula glutinis* Rh-00301	Fermenter (5 L)	Sucrose	Two-stage fed-batch (CF)	80	106	63	-	84	-	0.18	[27]
*Trichosporonoides spathulata* JU4-57	Fermenter (5 L)	Crude glycerol	One-stage fed-batch (IF)	144	17.3	7.25	-	-	-	-	[19]
			Two-stage fed-batch (IF)	132	13.8	7.78	-	-	-	-	
*Rhodosporidium toruloides* Y4	Fermenter (15 L)	Glucose	Fed-batch (IF)	146.7	89.0	52.2	-	36	-	0.20	[20]
			Fed-batch (CF)	138.5	127.5	78.8	-	57	-	0.23	
*Rhodotorula glutinis* BCRC21418	Fermenter (50 L)	Crude glycerol	Fed-batch (PF)	120	46.4	-	-	-	-	0.622	[28]
			Fed-batch (CF)	96	44.5	-	-	-	-	0.621	
			Fed-batch (EF)	84	39.2	-	-	-	-	0.433	

CF: Constant feeding mode or continuous feeding mode; IF: Intermittent feeding mode; PF: Pulse feeding mode; EF: Exponential feeding; -: No data.

**Table 4 microorganisms-08-00151-t004:** Fatty acid profiles of lipid produced by *Rhodosporidiobolus fluvialis* DMKU-SP314 using two-stage fed-batch cultivations with different feeding strategies.

Fatty Acid	Fatty Acid (% of Total Fatty Acid)							
	Batch Cultivation	Two-Stage Fed-Batch Cultivation						
		1st	2nd	3rd	4th	5th	6th	7th
Myristic acid (14:0)	1.0 ± 0.1 ^b^	1.1 ± 0.0 ^ab^	1.2 ± 0.0 ^ab^	1.6 ± 0.4 ^a^	1.6 ± 0.7 ^a^	1.1 ± 0.0 ^ab^	1.2 ± 0.1 ^ab^	1.1 ± 0.0 ^ab^
Palmitic acid (16:0)	29.2 ± 2.6 ^bc^	31.8 ± 0.4 ^a^	31.8 ± 0.6 ^a^	28.2 ± 0.3 ^c^	29.5 ± 0.5 ^bc^	31.1 ± 0.8 ^ab^	31.2 ± 1.6 ^ab^	29.7 ± 0.5 ^bc^
Palmitoleic acid (16:1)	0.7 ± 0.1 ^b^	0.6 ± 0.0 ^b^	0.6 ± 0.0 ^b^	0.7 ± 0.1 ^b^	0.7 ± 0.0 ^b^	0.6 ± 0.0 ^b^	1.1 ± 0.5 ^a^	0.9 ± 0.1 ^ab^
Stearic acid (18:0)	13.2 ± 0.2 ^a^	13.2 ± 0.2 ^a^	14.3 ± 0.8 ^a^	14.5 ± 0.3 ^a^	13.3 ± 0.0 ^a^	14.4 ± 0.5 ^a^	13.7 ± 0.5 ^a^	14.2 ± 1.2 ^a^
Oleic acid (18:1)	34.9 ± 0.3 ^a^	35.2 ± 0.2 ^a^	35.7 ± 0.3 ^a^	35.0 ± 0.4 ^a^	35.1 ± 0.3 ^a^	35.1 ± 1.0 ^a^	32.7 ± 0.5 ^b^	33.6 ± 0.2 ^b^
Linoleic acid (18:2)	17.3 ± 1.2 ^a^	15.2 ± 0.4 ^c^	13.5 ± 0.1 ^d^	15.9 ± 0.3 ^bc^	16.8 ± 0.0 ^ab^	14.9 ± 0.4 ^c^	16.5 ± 1.1 ^ab^	17.1 ± 0.8 ^ab^
γ-linolenic acid (18:3)	nd	nd	nd	nd	nd	nd	nd	nd
α-linolenic acid (18:3)	3.7 ± 0.9 ^ab^	2.9 ± 0.1 ^c^	2.9 ± 0.1 ^c^	4.2 ± 0.1 ^a^	3.1 ± 0.1 ^bc^	2.8 ± 0.4 ^c^	3.6 ± 0.1 ^ab^	3.5 ± 0.2 ^b^
Saturated fatty acids (SFAs)	43.4 ± 2.5 ^d^	46.1 ± 0.7 ^abc^	47.3 ± 0.2 ^a^	44.2 ± 0.2 ^cd^	44.4 ± 0.2 ^cd^	46.6 ± 1.2 ^ab^	46.1 ± 1.2 ^abc^	44.9 ± 0.8 ^bcd^
Unsaturated fatty acids (UFAs)	56.6 ± 2.5 ^a^	53.9 ± 0.7 ^bcd^	52.7 ± 0.2 ^d^	55.8 ± 0.2 ^ab^	55.6 ± 0.2 ^ab^	53.4 ± 1.2 ^cd^	53.9 ± 1.2 ^bcd^	55.1 ± 0.8 ^abc^

Cultivation were performed in a 3 L stirred tank-fermenter at 28 °C, 200 rpm agitation speed, and 2 vvm aeration rate; nd: Not detectable. Values in the same row which have different superscript letters were significantly different (*p* < 0.05).

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
