# Peer review of "Feeding Strategies of Two-Stage Fed-Batch Cultivation Processes for Microbial Lipid Production from Sugarcane Top Hydrolysate and Crude Glycerol by the Oleaginous Red Yeast Rhodosporidiobolus fluvialis"

_microorganisms, 2020, doi:10.3390/microorganisms8020151_

Round 1
Reviewer 1 Report
Dear Authors,
The work presented for evaluation is interesting and valuable. However, it requires corrections.
Specific comments:
line 33-34: "including bacteria, yeast, filamentous fungi, and 33 microalgae,..." - please list examples of microorganism species.
line: 48-49: "(formerly named 48 Rhodosporidium fluviale DMKU-SP314)" please delete. The current nomenclature for naming microorganisms should be used.
line 118-119: "Soybean powder was purchased from Doi Kham Food Products Co., Ltd, Thailand, which contained the total nitrogen content of 7%." Is this a manufacturer's declaration? Did the authors themselves determine the total nitrogen content? Such information should be given.
line 176-179:"The cell mass was determined gravimetrically and expressed as cell dry weight (CDW; g/L). Briefly, 5-10 mL of culture broth was centrifuged at 4,000 rpm, for 5 min, washed twice with distilled water, transferred to pre-weighed aluminum foil cup, and then dried at 80°C for 18-24 h before weight measurement was made."
The biomass determination method has a big mistake. For the future, please consider the optical density determination method. Of course, the optical density method also has its limitations. However, it is easier to standardize the OD method.
Figure 1, diagram B. Only 7.8,9 and 10 days should appear in the description under diagram B. The rest is unnecessary.
Table 1. No statistical analysis. What effect on glycerol concentration had on the synthesis of individual fatty acids. Please add statistical analysis.
Table 2. Please add statistical analysis.
Author Response
Response to Reviewer 1 Comments
Point 1: line 33-34: “including bacteria, yeast, filamentous fungi, and 33 microalgae,...” - please list examples of microorganism species.
Response 1: We added examples of microorganism species as “Oleaginous microorganisms are considered as oleaginous species that can accumulate intracellular lipids greater than 20% of their cell mass. For example, Nocardia globerula, Rhodococcus opacus and Streptomyces coelicolor, were reported as oleaginous bacteria. Oleaginous microalgae that were reported include, Chlorella protothecoides, C. vulgaris and Scenedesmus quadricauda. The most common oleaginous yeasts were Cryptococcus albidus, Lipomyces starkeyi, Rhodosporidium toruloides, Rhodotorula glutinis, and Yarrowia lipolytica. Moreover, some species of filamentous fungi viz. Aspergillus terreus, Mortierella alpina and M. isabellina, were reported to produce special lipids such as gamma-linolenic acid (GLA), arachidonic acid (ARA), eicosapentaenoic acid (EPA), and docosahexaenoic acid (DHA) [6,7].” in the revised manuscript (lines 37-45).
Point 2: line: 48-49: “(formerly named 48 Rhodosporidium fluviale DMKU-SP314)” please delete. The current nomenclature for naming microorganisms should be used.
Response 2: Deleted (lines 56).
Point 3: line 118-119: “Soybean powder was purchased from Doi Kham Food Products Co., Ltd, Thailand, which contained the total nitrogen content of 7%.” Is this a manufacturer’s declaration? Did the authors themselves determine the total nitrogen content? Such information should be given.
Response 3: The total nitrogen content of soybean powder was calculated from the protein content described in the manufacturer’s nutrition information. We did not determine by ourselves. Therefore, we revised this as “Soybean powder was purchased from Doi Kham Food Products Co., Ltd, Thailand, which contained 38.71% protein calculated to be the total nitrogen content of 7%.” (line 127).
Point 4: line 176-179: “The cell mass was determined gravimetrically and expressed as cell dry weight (CDW; g/L). Briefly, 5-10 mL of culture broth was centrifuged at 4,000 rpm, for 5 min, washed twice with distilled water, transferred to pre-weighed aluminum foil cup, and then dried at 80°C for 18-24 h before weight measurement was made.”
The biomass determination method has a big mistake. For the future, please consider the optical density determination method. Of course, the optical density method also has its limitations. However, it is easier to standardize the OD method.
Response 4: Thank you very much for your suggestion.
Point 5: Figure 1, diagram B. Only 7, 8, 9 and 10 days should appear in the description under diagram B. The rest is unnecessary.
Response 5: Corrected.
Point 6: Table 1. No statistical analysis. What effect on glycerol concentration had on the synthesis of individual fatty acids. Please add statistical analysis.
Response 6: Added the statistical analysis of data in Table 1-3 and added
“2.8 Statistical analysis
The statistical significance was evaluated by one-way analysis of variance (ANOVA) using IBM SPSS version 22 (SPSS, Cary, NC, USA) and the individual comparisons were obtained by Duncan’s Multiple Range Test (DMRT). A value of p < 0.05 was considered to indicate a significant difference between treatments.” in lines 199-203 of the revised manuscript.”
Point 7: Table 2. Please add statistical analysis.
Response 7: Added the statistical analysis in Table 2.

Reviewer 2 Report
Line 49:It states that "DMKU-SP314 was proved as potential lipid producer". Please mention the Citation# to reference this source. Not sure if this is in reference to [7] or [11], or both.
Line 60: Does "In our previous Study..." refer to [11], if so please state the same.
Line 111: What is the pitch of the shaker used in this study? Please mention the pitch of the shaker, if available.
Line 171: Please mention 'h', for 48, at the beginning of the line.
Line 177: I would recommend stating the true value of the volume of culture broth use to determine the CDW, as opposed to the range stated, as this help strengthen line 186-187, when represented as % value of fatty acids.
Line 248: For Fig 1A., the legend mentioned at the bottom for Day 2 is not the same as what is reflected in the figure.
Line 349: The line begins with a sentence that ends abruptly, "were."
Line 427: As stated that there needs to be further investigation to optimize the process, have the authors considered implementing temperature shifts during the second stage of fed batch cultivation, to increase production. This has been shown to be effective in certain mammalian cells and a few Oleaginous yeast strains. Something to incorporate in the next stage of process optimizations, using DOE.
Overall a well investigated and well written manuscript.
Author Response
Point 1: Line 49: It states that “DMKU-SP314 was proved as potential lipid producer”. Please mention the Citation# to reference this source. Not sure if this is in reference to [7] or [11], or both.
Response 1: Added the references [9,13] (line 57 of the revised manuscript).
Point 2: Line 60: Does “In our previous Study...” refer to [11], if so please state the same.
Response 2: Added the references [13] (line 70 of the revised manuscript).
Point 3: Line 111: What is the pitch of the shaker used in this study? Please mention the pitch of the shaker, if available.
Response 3: The shaker used in this study (TAITEC Bio-Shaker BR-300LF, Japan) is a horizontal orbital motion without pitch control.
Point 4: Line 171: Please mention “h”, for 48, at the beginning of the line.
Response 4: Edited (line 179 of the revised manuscript).
Point 5: Line 177: I would recommend stating the true value of the volume of culture broth use to determine the CDW, as opposed to the range stated, as this help strengthen line 186-187, when represented as % value of fatty acids.
Response 5: Edited (line 185 of the revised manuscript).
Point 6: Line 248: For Fig 1A., the legend mentioned at the bottom for Day 2 is not the same as what is reflected in the figure.
Response 6: Corrected. The symbols for Day 2 presented in Fig 1A look different from the legend because they have error bars.
Point 7: Line 349: The line begins with a sentence that ends abruptly, “were.”
Response 7: Deleted (line 363 of the revised manuscript).
Point 8: Line 427: As stated that there needs to be further investigation to optimize the process, have the authors considered implementing temperature shifts during the second stage of fed batch cultivation, to increase production. This has been shown to be effective in certain mammalian cells and a few Oleaginous yeast strains. Something to incorporate in the next stage of process optimizations, using DOE.
Response 8: Thank for your advice.
